# Investigation of the Microstructures and Properties of B-Bearing High-Speed Alloy Steel

**Jingqiang Zhang [1],*, Penghui Yang [2],* and Rong Wang [2]**

1   School of Mechanical and Energy Engineering, Zhejiang University of Science and Technology,
    Hangzhou 310023, China
2   State Key Lab Adv Proc & Recycling Nonferrous Met, Lanzhou University of Technology,
    Lanzhou 730050, China
*   Correspondence: jingqiangz@126.com (J.Z.); yangph@lut.edu.cn (P.Y.)

**Abstract:** This work aims to research the influence of boron and quenching temperature on the microstructures and performances of boron-bearing high-speed alloy steel. The results showed that the hardness and wear resistance of boron-bearing high-speed alloy steel were improved after increasing the boron content. The volume fraction of boron-rich carbide gradually decreased, and the hardness increased significantly with the rise in quenching temperature. The highest comprehensive mechanical properties were obtained for samples quenched at 1040 °C. The TEM results showed the boron-rich carbide was $M_7(C, B)_3$ with an HCP structure, and the precipitated particles were $M_{23}(C, B)_6$ with an FCC structure after tempering. This work may help improve the wear resistance of materials in the field of surface coatings.

**Keywords:** boron-bearing high-speed alloy steel; properties; quenching temperature; boron-rich carbide

## 1. Introduction

The roller is an important part of the production of hot rolling or cold rolling [1,2]. The working environment of the roller is very harsh, which requires good high-temperature oxidation resistance, high impact toughness, and excellent wear resistance [3,4]. High-speed alloy steel is often used in the roller due to its high hardness, good impact toughness, and excellent wear resistance [5,6]. However, a lot of alloying elements are also added to the high-speed alloy steel to enhance the wear resistance, which has increased the production cost, resulting in a huge waste of metallic resources. Surface coating technology is also often used to enhance the properties of a roller, but the wear resistance and the hardness of the substrate are also critical. Therefore, this work sets the scene for the remaining laser cladding works by studying the properties of the substrate.

To reduce the consumption of energy, low-cost materials are important to use in the equipment, particularly in wear-resistant materials. The element boron is often added to steel to replace expensive elements and improve wear resistance, as boron can combine with C, Fe, Cr, Mo, and other elements to form hard phases with good stability and high hardness. Moreover, the addition of boron can allow for its distribution into the matrix to enhance the hardenability of materials and, best of all, reduce the cost compared to other elements [7–9].

Chen et al. [8] researched the microstructures and properties of white cast iron with high boron content and found that $M_2B$-type eutectic compounds formed after adding boron. XRD, EDX, and TEM results showed that the $M'_{0.9}Cr_{1.1}B_{0.9}$ eutectic compounds appeared in the matrix of white cast irons when adding about 4 wt.% chromium. Fu et al. [9] investigated the microstructures, performance, and wear resistance of boron-bearing high-speed steel. They found that the casting-state structure of boron-bearing high-speed steel consisted of $M_2(B,C)$ carbides, $M_{23}(B,C)_6$, α-Fe, $M_3(B_{0.7}C_{0.3})$, TiC, and a small quantity of retained austenite. The impact toughness and hardness reached 80–85 kJ/cm² and 65–67

HRC, respectively. However, this high-speed steel with low carbon and high alloy content did not achieve high wear resistance, and the W and Ti elements were expensive.

Therefore, in this work, boron will be added to high-speed alloy steel with high carbon content to study the influence of boron and heat treatment processes on high-speed alloy steel. More importantly, the developed high-speed alloy steel will be used to investigate if the use of expensive elements could be decreased by adding boron to reduce costs.

## 2. Experiment Details

### 2.1. Material Preparation

The boron-bearing high-speed alloy steel was prepared in a 40 kg medium-frequency induction furnace. All raw materials were melted, and ferroboron was added into the melt as the temperature reached 1540 °C. The melt was poured into sand molds when the temperature rose to 1610 °C, and then *Y*-block ingots were obtained according to the standard of ASTM A781/A 781-M95. Table 1 shows the chemical constituent of the testing alloy steel.

**Table 1.** The chemical constituent of the testing alloy steel (wt.%).

| Alloy | C | B | Mo | Si | Mn | Cr | V | S | P | Fe |
|-------|------|------|------|------|------|------|------|-------|-------|------|
| A | 1.19 | 1.03 | 1.36 | 0.87 | 1.13 | 3.50 | 1.13 | 0.011 | 0.024 | Bal. |
| B | 1.23 | 1.59 | 1.41 | 0.92 | 1.24 | 3.48 | 1.05 | 0.016 | 0.025 | Bal. |

The heat treatment processes of boron-bearing high-speed alloy steel were as follows: the alloy steel was austenitized at different temperatures for 1 h, then quenched by air. After that, the samples were tempered at 500 °C for 3 h and then cooled by air.

### 2.2. Microstructural Investigation

The microstructures of the samples were investigated by a scanning electron microscope (SEM, JSM-6510, JEOL, Tokyo, Japan) (secondary electron detector), an optical microscope (OM, Olympus BX51, Olympus, Tokyo, Japan), an energy dispersive spectroscope (EDS, Oxford Instruments, Oxford, London, England) with an acceleration voltage of 15 kV, a transmission electron microscope (TEM, JEM-2100F, JEOL, Tokyo, Japan) [10] with an energy-dispersive X-ray (EDX, Oxford Instruments, 80 mm$^2$ X-max SDD, Oxford, London, England, and an electron probe X-ray micro-analyzer (EPMA, JXA-8230, EOL, Tokyo, Japan). The samples were polished and then etched by a 5 g $CuSO_4$ + 20 mL HCl + 20 mL $H_2O$ solution for metallographic observation. The volume fraction of the boron-rich carbide (B-rich carbide) was measured by Image-Pro software (1.0, Tokyo, Japan). Fifty different fields of view were selected and calculated the average value was the final result (the standard deviation was less than 4). The phase composition was analyzed by X-ray diffraction (XRD, Rigaku D/Max-2400× diffractometer, Rigaku, Tokyo, Japan) with a 2θ range of 10°–90° and a scanning speed of 20°/min.

### 2.3. Properties Investigation

The hardness of boron-bearing high-speed alloy steel was measured by the HR-150A Rockwell hardness tester (TIME, Beijing, China), and the average value was calculated as the resulting value (the standard deviation was less than 5). The microhardness measurements of boron-bearing high-speed alloy steel samples were carried out by a MICRO MET-5103 Vickers hardness tester with a load of 1.96 N. According to the standard of ASTM E384-1999, the unnotched Charpy impact specimens were measured by a JBW-300 capacity impact testing machine with an impact energy of 150 J at room temperature. The size of the samples for the impact test was 10 mm × 10 mm × 55 mm. Wear loss was measured by an MM-200 block-on-ring wear testing machine with a wheel rotation speed of 300 r/min, a load of 300 N, and a test duration of 60 min. The size of the specimen for the wear test was 10 mm × 10 mm × 10 mm. The grinding ring with a material of GCr15 steel, a hardness

of 60 HRC, and a size of φ40 mm × 10 mm (thickness) was selected to rub the specimen for the wear test. In order to compare wear resistance, the wear loss was measured by the electronic balance (TG328B), and then the average value of three samples was calculated as the final weight loss result. The worn surfaces of the specimens were analyzed by a VK-9710 color 3D laser scanning microscope (Keyence, Osaka, Japan) and SEM.

## 3. Results and Discussion

### 3.1. As-Cast Microstructures

Figure 1 shows the as-cast microstructures of the boron-bearing high-speed alloy steel (B-bearing high-speed alloy steel). The solidification microstructures of the two boron-bearing high-speed alloy steel are mainly constituted of the matrix and the coarse net-like B-rich carbides. Thereinto, the matrix contains ferrite mainly, some pearlites, and less martensite. The area of B-rich carbide was calculated by Image-Pro software (Version 6.0), and the results showed that the volume fraction of B-rich carbide in alloy A and alloy B was 23.1 and 28.9 vol.%, respectively. Evidently, the quantity of B-rich carbide in alloy B was more than in alloy A, indicating that the B-rich carbides increased with the increase in boron content.

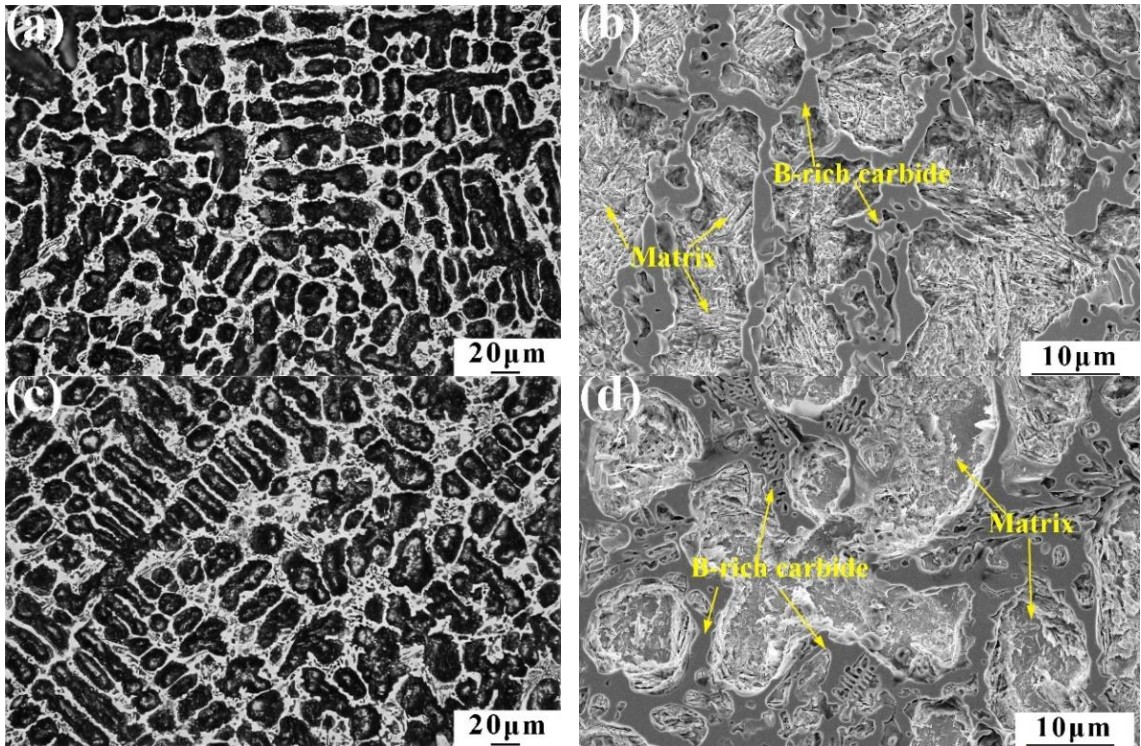

**Figure 1.** Microstructures of alloy steel as-cast: (**a**) OM result of alloy A; (**b**) SEM result of alloy A; (**c**) OM result of alloy B; (**d**) SEM result of alloy B.

In order to study the element constitution and the distribution of B-rich carbide in the two alloys, the surface of specimens was observed by an SEM with an EDS. Figures 2 and 3 show the surface scanning results of alloys A and B, respectively. Table 2 shows EPMA test results of B-rich carbides. The results showed that the elements contained in the carbides are C, B, Cr, Mo, etc., indicating that the Cr and Mo elements were dissolved in the B-rich carbide. The microhardness of the B-rich carbide in alloy A and alloy B is 644 and 731 $HV_{0.2}$, which indicates the hardness of B-rich carbide increased with B content. In addition, the microhardness of the matrix in alloy A and alloy B is 255 and 246 $HV_{0.2}$, indicating that the matrix hardness did not significantly change after increasing the B content.

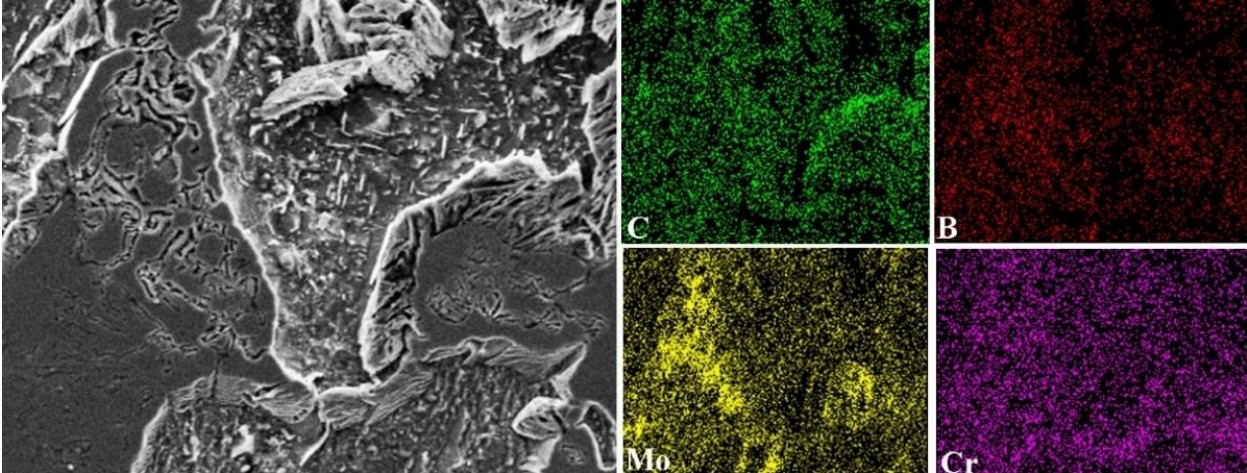

**Figure 2.** EDS analysis of as-cast alloy A.

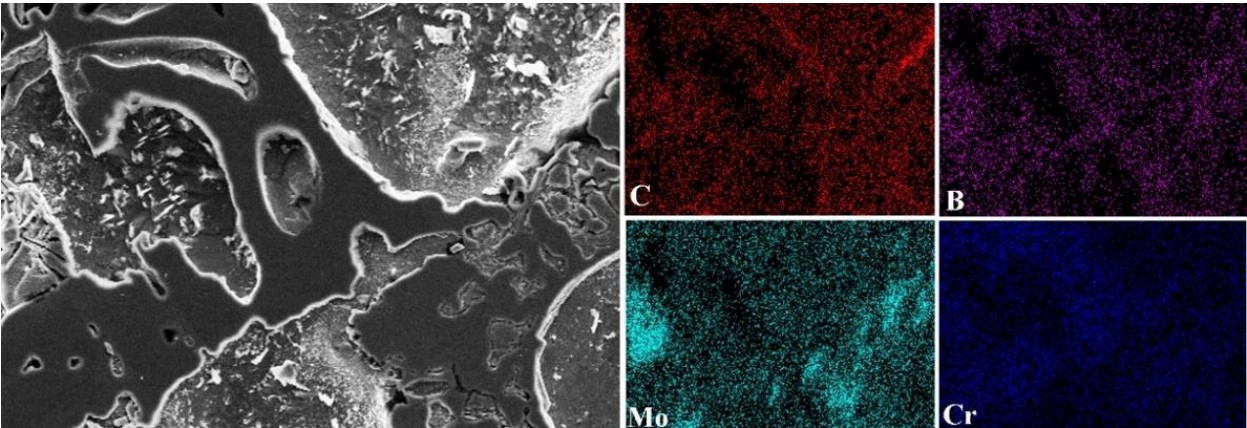

**Figure 3.** EDS analysis of as-cast alloy B.

**Table 2.** EPMA test results of B-rich carbides (at.%).

| Alloy | C | B | Mo | Si | Mn | Cr | V | Fe |
|---|---|---|---|---|---|---|---|---|
| A | 13.45 | 18.33 | 2.88 | 0.96 | 0.24 | 6.77 | 1.79 | 55.58 |
| B | 11.26 | 22.69 | 3.15 | 0.63 | 0.51 | 5.24 | 1.81 | 54.71 |

### 3.2. Effect of Quenching Temperature on Microstructures and Properties

Figure 4 shows the microstructure of alloy A after quenching at different temperatures. After quenched at 960 and 1000 °C, the matrix of the boron-bearing high-speed alloy steel contained ferrite and less pearlite, as shown in Figure 4a–d. The matrix changed from ferrite to martensite when the quenching temperature exceeded 1040 °C (see Figure 4e,f). With the increase in quenching temperature, the matrix did not change, as shown in Figure 4a–d, indicating that the high quenching temperature could obtain the martensitic matrix.

Figure 5 shows the microstructure of alloy B after quenching at different temperatures. Similar to the case of alloy A, the matrix of boron-bearing high-speed alloy steel contained ferrite and less pearlite after quenching at relatively low quenching temperatures, as shown in Figure 5a–d. After quenching at relatively high quenching temperatures (more than 1040 °C), the matrix of the boron-bearing high-speed alloy steel was mainly composed of martensite.

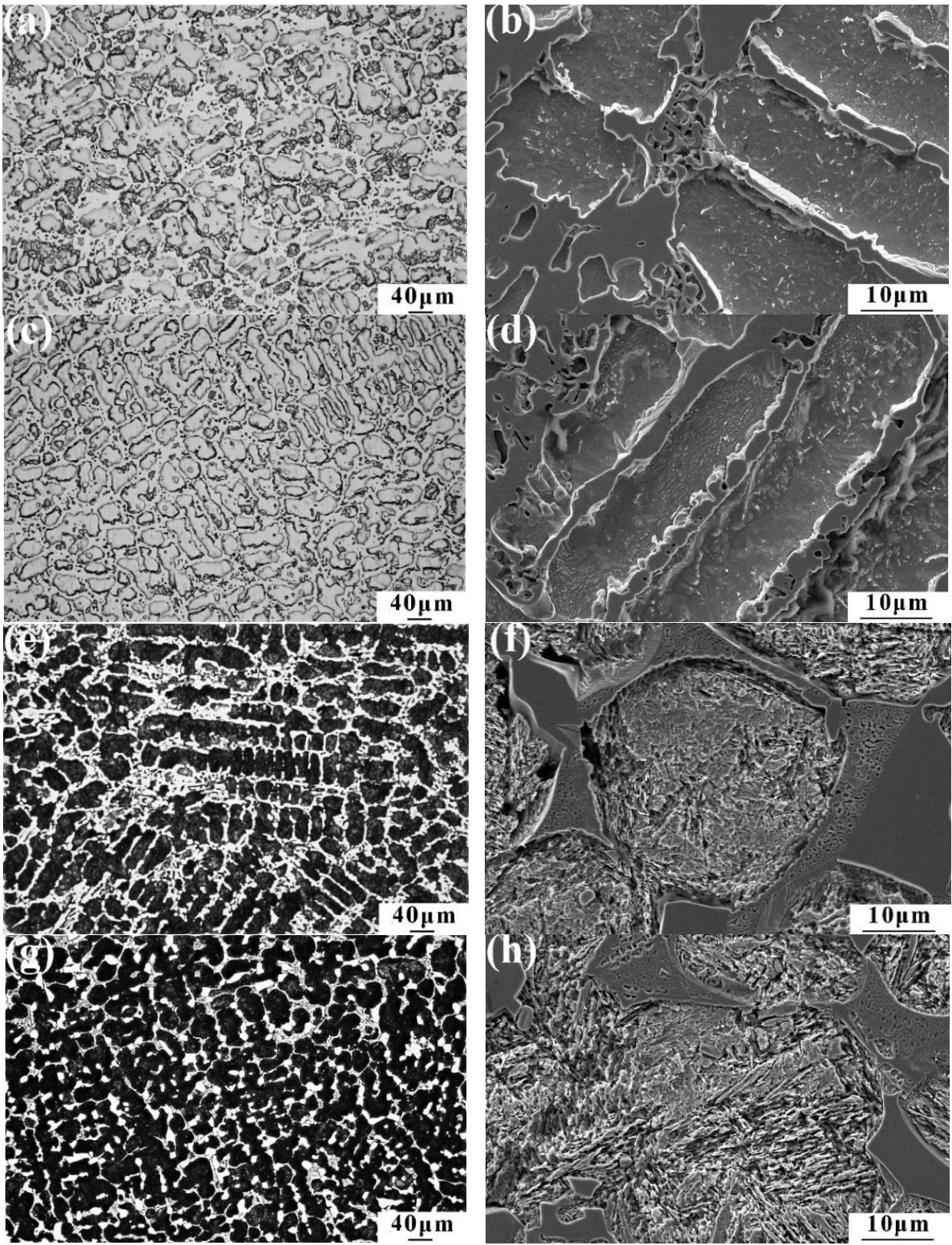

**Figure 4.** Microstructures of alloy A quenched at different temperature (°C): (**a**,**b**) 960; (**c**,**d**) 1000; (**e**,**f**) 1040; (**g**,**h**) 1080.

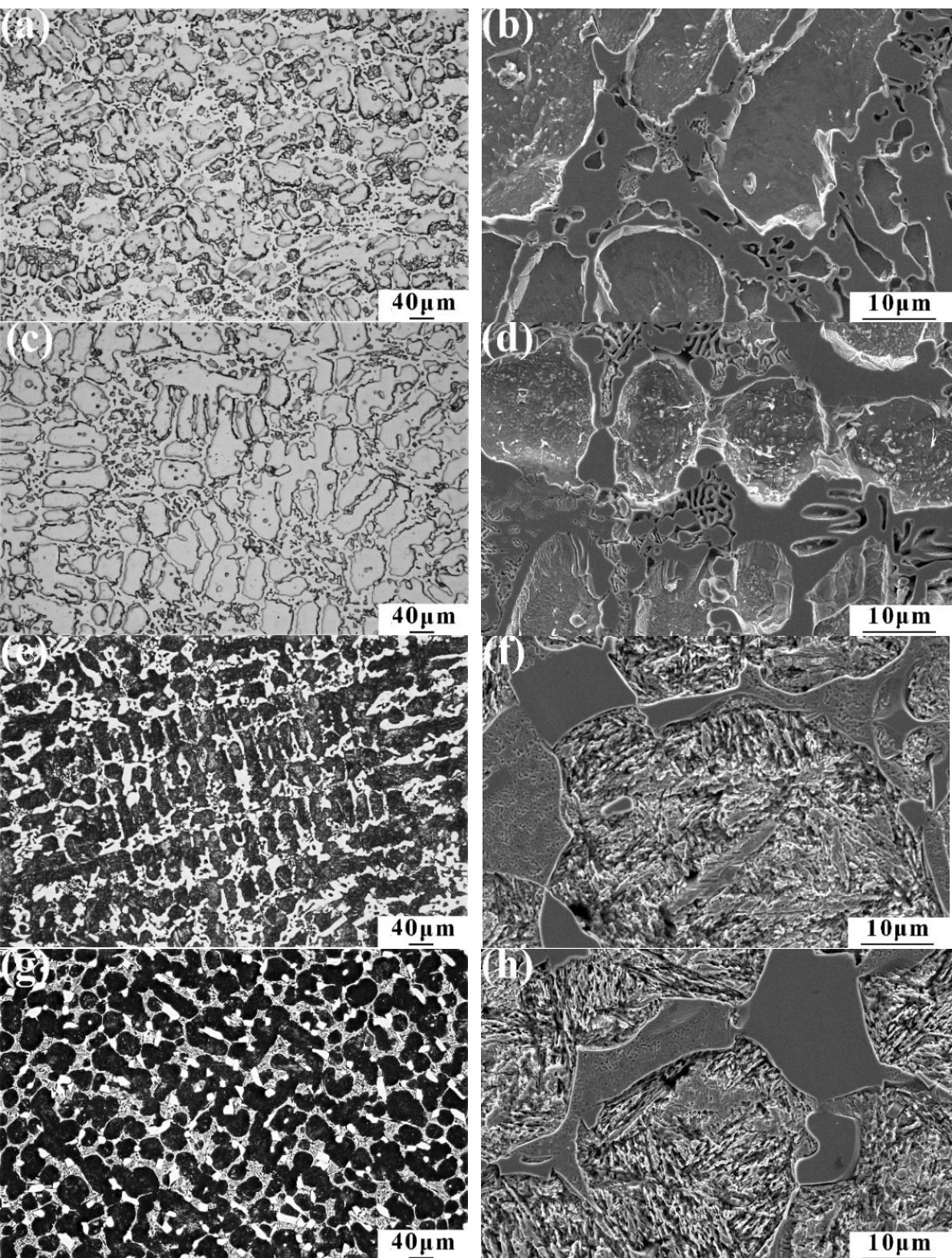

**Figure 5.** Microstructures of alloy B after quenching at different temperature (°C): (**a**,**b**) 960; (**c**,**d**) 1000; (**e**,**f**) 1040; (**g**,**h**) 1080.

To further analyze the influence of quenching temperature on B-rich carbide, the volume fraction of B-rich carbide in B-bearing high-speed alloy steel quenched at different

temperatures was investigated. The computed results are shown in Figure 6. As the quenching temperature increased, the volume fraction of B-rich carbide gradually decreased. Alloy A and alloy B showed the same trend. High temperatures promoted the increase of boron solid solubility in austenite, which caused the elements in the B-rich carbide to diffuse into the matrix, resulting in the dissolution of the B-rich carbide. It is also worth noting that the volume fraction of B-rich carbide in alloy A was always smaller than in alloy B.

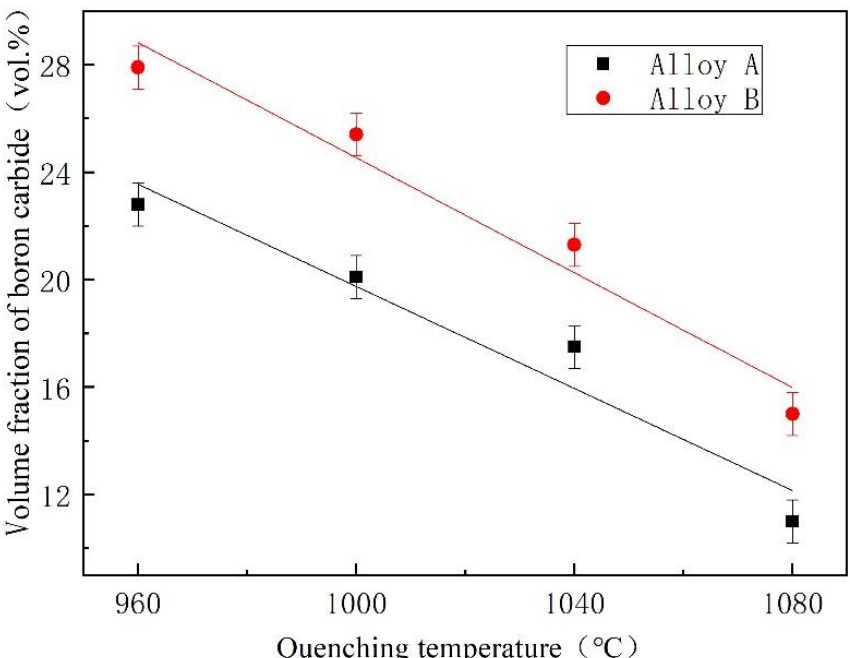

**Figure 6.** The volume fraction of B-rich carbide in boron-bearing high-speed alloy steel quenched at different temperatures.

Figure 7 shows the influence of quenching temperature on microhardness in the alloy steel matrix. The microhardness of the matrix of alloy A and alloy B increased significantly with the increase in quenching temperature. When quenched at 1080 °C, the microhardness of alloy A and alloy B reached the highest values, 720 and 780 $HV_{0.2}$, respectively. When the quenching temperature was relatively low, the matrix of the boron-bearing high-speed alloy steel was mainly composed of ferrite. After the quenching temperature reached 1080 °C, the ferrite matrix became martensite, which improved the hardness of the alloy. Furthermore, the elements in B-rich carbide dissolved into the matrix, which led to an increase in matrix hardness.

Figure 8 shows the properties of the boron-bearing high-speed alloy steel after quenching at different temperatures. The macro-hardness trend was similar to that of microhardness. Although the increase in the quenching temperature led to a decrease in B-rich carbide, the increase in matrix hardness promoted an increase in microhardness. The impact toughness gradually increased with the rise in quenching temperature. The structure of the boron-bearing high-speed alloy steel contained ferrite, less pearlite, and a large volume fraction of carbides at the low quenching temperature, so the hardness and impact toughness were lower. The appearance of martensite promoted an increase in hardness, and a decrease in carbides promoted an increase in impact toughness. However, the appearance of a large amount of martensite could destroy the impact toughness. For the two alloys, the quenching temperature to obtain the optimal comprehensive mechanical properties was 1040 °C. Therefore, alloys A and B were treated with a quenching temperature of 1040 °C to study the microstructure and wear resistance after tempering.

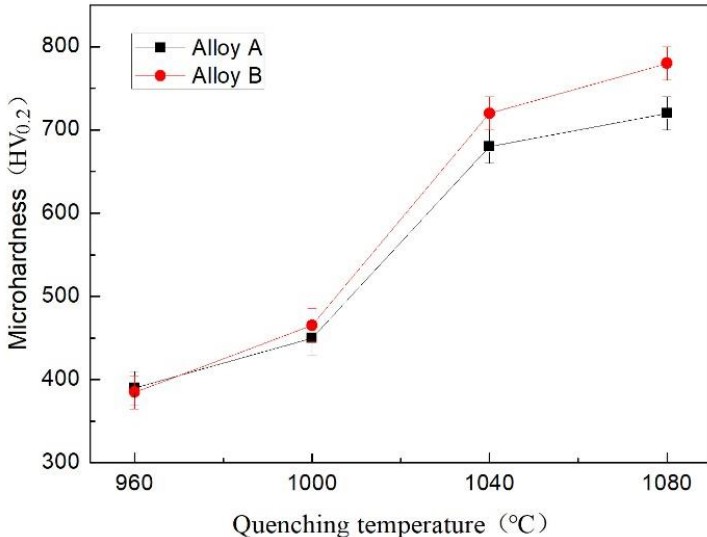

**Figure 7.** The microhardness in matrix of alloy steel.

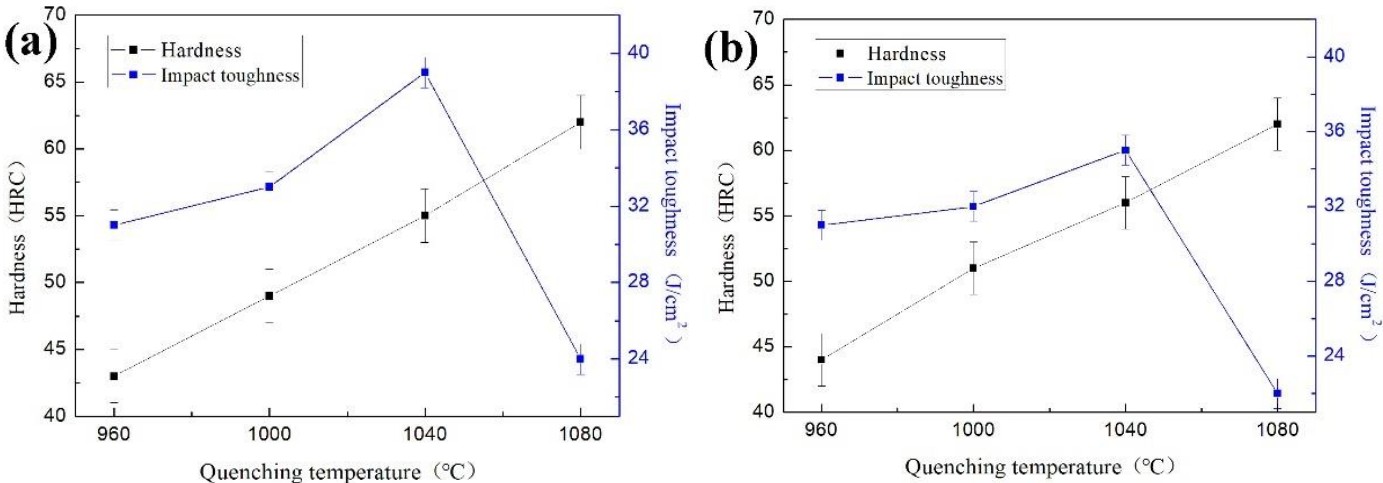

**Figure 8.** Hardness and impact toughness of alloy steel: (**a**) alloy A; (**b**) alloy B.

*3.3. Microstructure and Wear Resistance after Tempering*

　　Figure 9 shows the microstructures of alloys A and B in the heat-treated condition. The microstructure of the B-bearing high-speed alloy steel comprises precipitated particles, martensite matrix, and B-rich carbide after tempering at 500 °C for 3 h. It is worth noting that the number of particles was relatively small. The total volume fraction of B-rich carbide and precipitated particles were investigated by Image-Pro software, and the total volume fraction of alloy A and B were 19.2 and 23.3 vol.%, respectively, indicating that the total volume fraction of alloy A was slightly less than alloy B.

　　In order to further analyze the types of B-rich carbides and precipitated particles, the microstructures of alloy B were investigated by a TEM with an EDX. The results of the TEM micrographs and elemental analysis are shown in Figure 10 and Table 3, respectively. The B-rich carbide is $M_7(C, B)_3$ with a hexagonal close-packed (HCP) structure [11], and the lattice constants are a = 1.38 nm and b = 0.43 nm, as shown in Figure 10a. The precipitated particles are $M_{23}(C, B)_6$ with a face-centered cubic (FCC) structure [11], and the lattice constant is a = 1.07 nm, as shown in Figure 10b. Figure 11 shows the XRD results of the boron-bearing high-speed alloy steel after tempering. The matrix contained two kinds of carbides: $M_{23}(C, B)_6$ and $M_7(C, B)_3$, which is in agreement with the TEM results. In addition, the results of the high-resolution transmission electron microscopy (HR-TEM) in Figure 10c also provided support.

**Figure 9.** Microstructure of boron-bearing high-speed alloy steel after tempering: OM results (**a**), SEM results (**c**) and SEM results at high magnification (**e**) of alloy A; OM results (**b**), SEM results (**d**) and SEM results at high magnification (**f**) of alloy B.

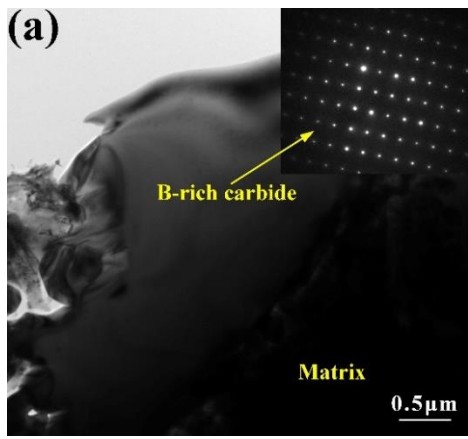
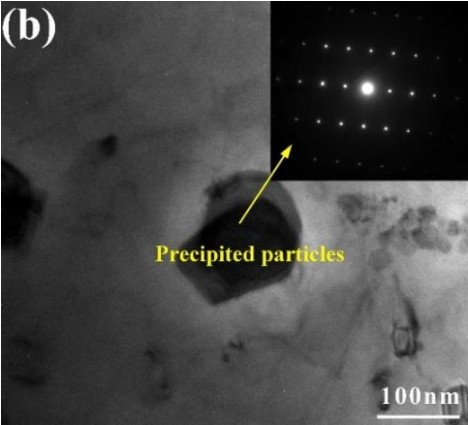

**Figure 10.** *Cont*.

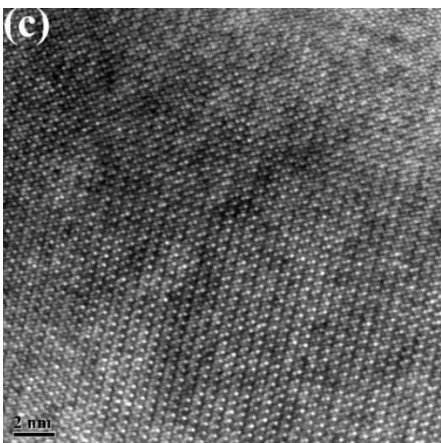

**Figure 10.** Bright-field TEM micrographs and corresponding selected area diffraction patterns (SADPs) of B-rich carbide and precipitated particles: (**a**) bright-field TEM micrographs and SADPs of B-rich carbide; (**b**) bright-field TEM micrographs and SADPs of precipitated particles; (**c**) HR-TEM image of precipitated particles.

**Table 3.** EDX test results of TEM samples (at%).

| Phase | C | B | Mo | Si | Mn | Cr | V | Fe |
|---|---|---|---|---|---|---|---|---|
| B-rich carbides | 11.68 | 19.65 | 3.56 | 1.12 | 0.56 | 5.69 | 1.52 | 56.13 |
| Precipited particles | 9.88 | 13.21 | 2.64 | 1.35 | 0.94 | 6.99 | 2.56 | 62.43 |

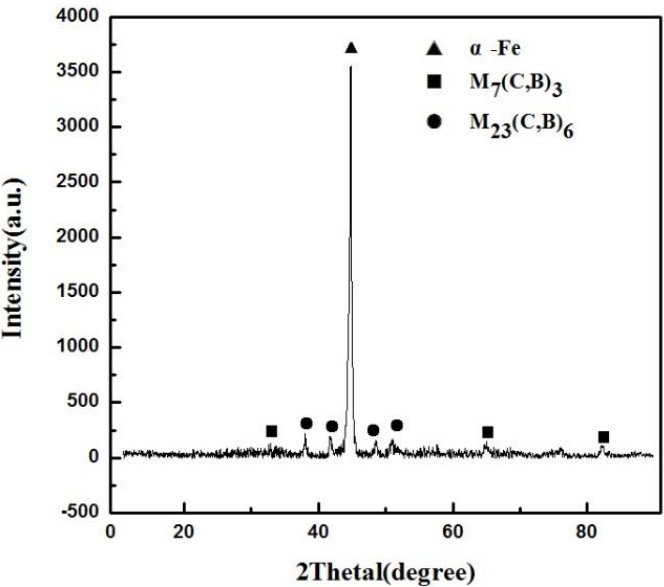

**Figure 11.** XRD result of B-bearing high-speed alloy steel after tempering.

The wear resistance of alloys A and B was researched by a wear tester, and the reciprocal of the wear loss was expressed as the wear resistance. The wear loss of alloys A and B are shown in Figure 12. The results show the wear resistance of alloy B exceeded alloy A, which indicates that the wear resistance of boron-bearing high-speed alloy steel increased after increasing the B content. The increase in wear resistance is mainly attributed to the increase in the volume fraction of carbides.

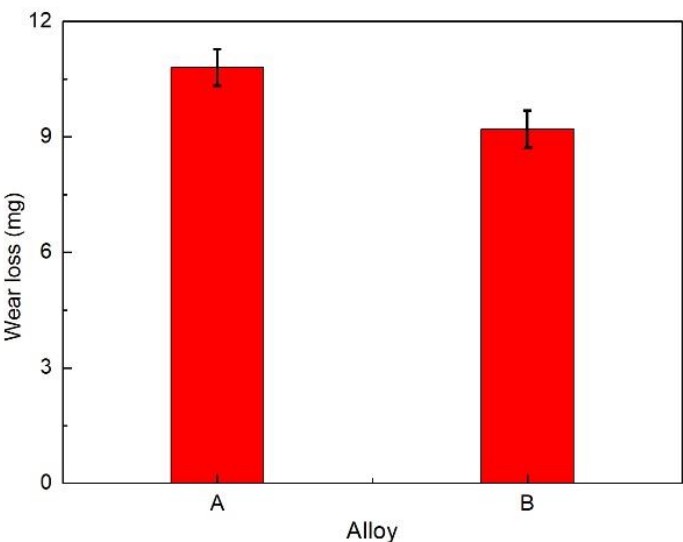

**Figure 12.** Wear loss of alloy A and alloy B.

The worn surface of the alloy steel was observed by an SEM, as shown in Figure 13a,b. The spalling and groove could be observed on the worn surface of the boron-bearing high-speed alloy steel, indicating that the wear mechanism of boron-bearing high-speed alloy steel may be abrasive wear. To reveal the wear mechanism of boron-bearing high-speed alloy steel, the cross-section of alloys A and B after wear was observed by an SEM, see Figure 13c,d. The B-rich carbide could protect the matrix, indicating that the wear resistance is significantly enhanced after increasing the B content.

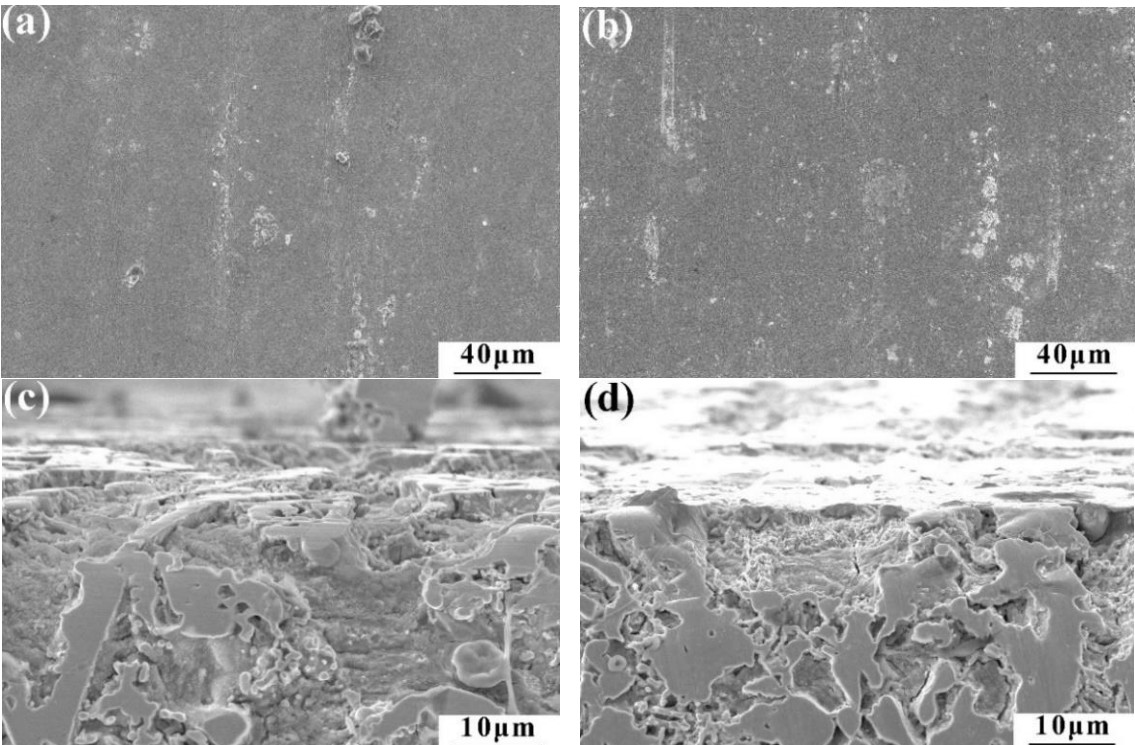

**Figure 13.** SEM images of B-bearing high-speed alloy steel worn surfaces: (**a**) SEM images of alloy A worn surfaces; (**b**) SEM images of alloy B worn surfaces; (**c**) SEM image of cross-section of alloy A after wear; (**d**) SEM image of cross-section of alloy B after wear.

To analyze the wear resistance of B-bearing high-speed alloy steel, the worn surface was observed by a 3D laser scanning microscope, see Figure 14. Due to the existence of B-rich carbide, the shedding was easily formed during the process of wear. Additionally, shallow grooves and low roughness occurred on the surface of alloy B because the wear resistance of alloy B is better than alloy A.

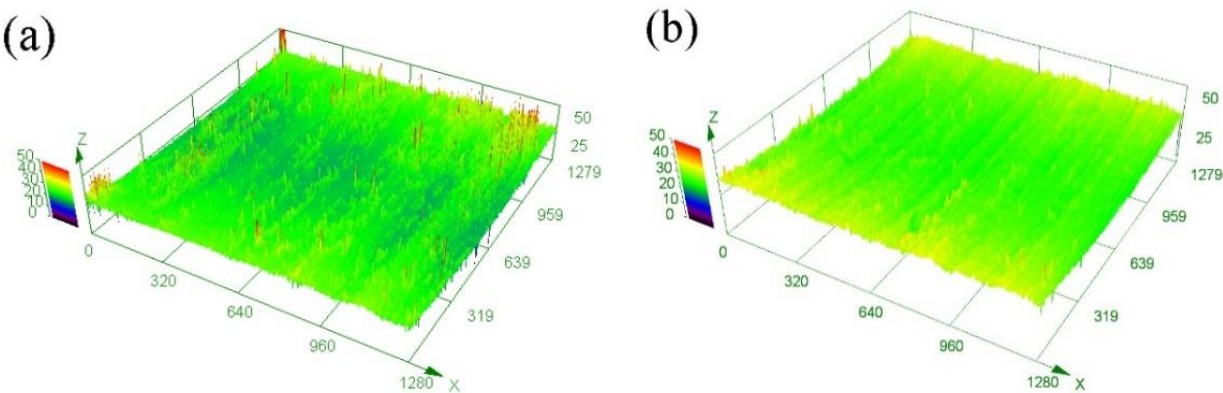

**Figure 14.** 3D laser morphologies of B-bearing high-speed alloy steel: (**a**) alloy A; (**b**) alloy B.

## 4. Conclusions

In this paper, boron was added to high-speed alloy steel with high carbon content to study the effects of boron and heat treatment on high-speed alloy steel. The conclusions obtained are as follows:

(1)  With the increase in austenitizing temperature, the quantity of B-rich carbide gradually decreased. The comprehensive mechanical properties were optimal after the alloy steel was austenitized at 1040 °C for 1 h, quenched in air, and tempered at 500 °C for 3 h.

(2)  TEM results showed the B-rich carbide is $M_7(C, B)_3$ with an HCP structure, and the precipitated particles are $M_{23}(C, B)_6$ with an FCC structure after tempering.

(3)  The wear resistance of boron-bearing high-speed alloy steel was enhanced by increased B content.

**Author Contributions:** Conceptualization, J.Z. and P.Y.; methodology, J.Z.; software, R.W.; investigation, R.W. and P.Y. All authors have read and agreed to the published version of the manuscript.

**Funding:** This research received no external funding.

**Institutional Review Board Statement:** Not applicable.

**Informed Consent Statement:** Not applicable.

**Data Availability Statement:** The datasets generated during and/or analyzed during the current study are available from the corresponding author on reasonable request.

**Conflicts of Interest:** The authors declare no conflict of interest.

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
