# Peer review of "Investigation of the Microstructures and Properties of B-Bearing High-Speed Alloy Steel"

_coatings, doi:10.3390/coatings12111650_

Round 1

Reviewer 1 Report

The authors present a study of boron containing tool steel in which the effects of boron content and austenitizing temperature are studied. After determining an appropriate austenitizing temperature, the wear performance is investigated. Overall, the manuscript provides insufficient detail on the experimental methods and fails to identify clear structure-property relationships to explain the changes in behavior. A number, of specific comments are included below.

-It is not appropriate to describe the boride phase as boron carbide. Instead, these appear to be typical metallic carbides with B substitution for C. However, additional work is necessary to confirm the composition. The EDS results as presented are insufficient; EDS should be performed on the TEM samples to isolate the carbide phases.

-More detail on the SEM-based EDS measurements is necessary. What type of detector was used? What was the accelerating voltage? The elemental maps provided do not show an apparent correlation between chemistry and morphology, which could be due to a number of factors. Inclusion of point spectra could aid in increasing confidence in the EDS measurements. Finally, maintaining a consistent color scheme between the elemental maps would aid the reader in understanding the results.

-More details on the wear measurement are necessary. What was the counter surface? What was the sliding distance? The wear volume should be measured in addition to the mass loss.

-How were the austenitizing temperatures selected? Are the samples fully austenitized at all temperatures?

-The methods explain some XRD measurements. However, no XRD measurements are included. XRD measurements would aid in identifying the phases present in the material.

-What was the applied load for the Vickers microhardness? Is the plastic zone truly contained within a single phase?

Finally, the manuscript is not appropriate for the special issue. The study does not pertain to laser cladding beyond the brief suggestions that the B-containing steel could be a laser cladding substrate.

Author Response

The authors appreciate the helpful comments provided by the referee. The revisions addressing the issues and comments are listed and explained below, in addition to those revisions and corrections from our own considerations. To facilitate the review process, response to each of the issues and comments is presented item by item. Moreover, thank you for your suggestion very much, and we benefit greatly from these suggestions.

Reviewer 2 Report

The author presented the high-speed alloy steel with added boron. The manuscript is interesting and well organized. However, there are major shortcomings. The following comments are:

1.        Page 2 lines 73 – 75. It should be shown how large the standard deviation was. Is it negligible?

2.        Page 2 lines 77 – 79. It should be shown how large the standard deviation was. Is it negligible?

3.        Page 2 line 84. What is TG328B?

4.        Chapter 2.1 Is it similar material to alloy A or B? Is it any normative chemical composition of the material without B? What was the motivation to use such kind a chemical composition?

5.        Fig. 6 A regression line should be estimated. This allows comparing the change of the volume fraction of boron carbine with the temperature. Additionally, a regression line with two parameters could answer, is the slope coefficient the same for alloys A and B?

6.        In conclusion, there are no clear recommendations for heat treatment. Giving only the quenching temperature is not enough.

7.        From the manuscript doesn’t appear that adding boron gives better properties. The comparison with similar material without boron must be given (own results or from literature). Additionally, the material has reached a lower value of hardness and impact toughness than mentioned HSS in lines 49-50.

Additional, there are some particular comments, as follows:

Fig. 11 scatter bands are hardly visible.

Figures and legends should be on the same page.

Author Response

Thank you very much for your attention and comments on this paper. I have learned a lot from your guidance. We have revised the manuscript according to your advices and suggestions below.

Reviewer 3 Report

The paper shows investigations on microstructures and properties of B-bearing high-speed alloy steel. Some changes need to be made in order to be published. Following are the changes:

(1) Rolls is not a process. Please change the introduction.

(2) The experimental procedure lacks important information, such as the procedure adopted for SEM and TEM preparation as EDS information.

(3) On the hardness part, more information is also lacking.

(4) Until the results, the authors use B, then use it in full.

(5) When presenting hardness, authors must present the value according to the corresponding standard.

(6) In the experimental procedure, the authors say that they used a hardness method but later present another method in the results.

(7) In Figure 10, the SAED images must be indexed.

(8) In Figure 13, there is not much difference between the two images.

Author Response

(The authors gave the same response as above.)

Round 2

Reviewer 1 Report

The authors have considerably improved the manuscript in by addressing some of the original reviewer comments. However, several of the original critiques remain insufficiently addressed.

-The selection of the austenitizating temperatures remains unexplained. The provided phase diagram in the response letter is helpful but insufficient. Was there an expected benefit to austenitizing in the ferrite+austenite phase field?

-The presentation of the EDS measurement methods and data are still lacking. For instance, what EDS detector type was used? Both EDS and EDX are used; they are the same technique so choose one acronym. It is unsurprising that distinguishing B and C are difficult hence the question about the detector type. Given the difficulty of quantifying light elements how is the reader to have any confidence in the results as presented? The use of EPMA is a good choice but again the experimental details are lacking.

-For such high volume fractions reported from the microscopy it is surprising how low the carbide fraction appears to be from the XRD. Is there any explanation for the discrepancy?

Finally, little has been done to more clearly identify a structure-property relationship to explain the various behaviors. Combined with the poor tie in to the scope of the special issue, the reviewers opinion that the manuscript is inappropriate for the special issue on "Laser Cladding Coatings" remains unchanged.

Author Response

Thank you very much for your attention and comments on this paper. In addition, thank you very much for your approval of our modification. I have learned a lot from your guidance. We have revised the manuscript according to your advices and suggestions below.

Reviewer 2 Report

I accept manuscript in present form.

Author Response

Thank you so much

Reviewer 3 Report

The paper can be now accepted. 

Author Response

Thank you so much

Round 3

Reviewer 1 Report

The authors continue to make many improvements to the manuscript. Most of the reviewer comments have been adequately addressed.

However, the authors have not satisfactorily addressed the appropriateness of the manuscript for the special issue on Laser Cladding. I cannot endorse publication in this special issue as the content does not fall within the scope. I would support publication as either a general article in Coatings or in another MPDI journal such as Metals or Materials.